# Antimicrobial Resistance in Physiological and Potentially Pathogenic Bacteria Isolated in Southern Italian Bats

**DOI:** 10.3390/ani13060966

**Published:** 2023-03-07

**Authors:** Maria Foti, Rosario Grasso, Vittorio Fisichella, Antonietta Mascetti, Marco Colnaghi, Maria Grasso, Maria Teresa Spena

**Affiliations:** 1Department of Veterinary Science, University of Messina, Via Palatucci 13, 98168 Messina, Italy; 2Department of Biological, Geological and Environmental Sciences, University of Catania, Via Androne 81, 95124 Catania, Italy; 3Department of Experimental and Applied Psychology, Vrije Universiteit Amsterdam, Van der Boechorststraat 7, 1081 HV Amsterdam, The Netherlands

**Keywords:** bats, antibiotic resistance, Gram-negative bacteria, Gram-positive bacteria, disk diffusion test

## Abstract

**Simple Summary:**

One of the most relevant health threats in recent years has been the antimicrobial resistance of both pathogenic and commensal bacteria. The aim of the study was to evaluate the levels of drug resistance among 413 Gram-negative and 183 Gram-positive bacteria, previously isolated from six bat populations living in Sicilian and Calabrian territory (Italy), using the disk diffusion method. Antimicrobial susceptibility analysis showed high resistance to some of the molecules tested and the presence of numerous multi-drug-resistant (MDR) strains.

**Abstract:**

The spread of antimicrobial resistance is one of the major health emergencies of recent decades. Antimicrobial-resistant bacteria threaten not only humans but also populations of domestic and wild animals. The purpose of this study was to evaluate the distribution of antibiotic resistance (AMR) and multidrug resistance (MDR) in bacterial strains isolated from six Southern-Italian bat populations. Using the disk diffusion method, we evaluated the antimicrobial susceptibility of 413 strains of Gram-negative bacteria and 183 strains of Gram-positive bacteria isolated from rectal (R), oral (O) and conjunctival (C) swabs of 189 bats belonging to 4 insectivorous species (*Myotis capaccinii*, *Myotis myotis*, *Miniopterus schreibersii* and *Rhinolophus hipposideros*). In all bat species and locations, numerous bacterial strains showed high AMR levels for some of the molecules tested. In both Gram-negative and Gram-positive strains, the resistance patterns ranged from one to thirteen. MDR patterns varied significantly across sites, with Grotta dei Pipistrelli in Pantalica displaying the highest levels of MDR (77.2% of isolates). No significant differences were found across different bat species. Monitoring antibiotic resistance in wildlife is a useful method of evaluating the impact of anthropic pressure and environmental pollution. Our analysis reveals that anthropic contamination may have contributed to the spread of the antibiotic resistance phenomenon among the subjects we examined.

## 1. Introduction

The recent COVID-19 pandemic highlighted the potential role of bats as a reservoir and vehicle for dangerous infectious agents, underscoring the importance of implementing surveillance and monitoring strategies. While a wide number of human pathogenic viruses have been isolated from wild bat populations [1,2,3,4], less attention has been paid to bacterial agents [5,6]. In particular, little information is available concerning the occurrence of antimicrobial-resistant bacteria in wild bat populations.

In recent decades, the spread of antimicrobial resistance (AMR) has emerged as a major health emergency, with over a million deaths directly attributable to AMR in 2019 alone [7] and a tragic increase in treatment failures. In fact, when a new antibiotic molecule is introduced and begins to be employed in therapy circuits, the appearance of strains resistant to that molecule is typically recorded within a short period of time. Resistance therefore continuously conditions the choice, use, and future development of antibiotics. For these reasons, the phenomenon of antibiotic resistance is profoundly relevant to, and has a strong impact on, clinical practice. The spread of AMR has been linked to the excessive and often indiscriminate use of antimicrobials in human and veterinary medical practice, as well as in agricultural farming practices.

Besides humans and domestic animals, antimicrobial-resistant bacteria are often present in wild animal populations, which therefore play a major role in the diffusion of AMR [8,9,10]. Wild animals can acquire resistant bacteria from the environment, especially through contaminated food or water, and can therefore be considered indicators of environmental pollution from resistant bacteria or genetic determinants of resistance [11]. The spread of resistant bacterial strains is strongly linked to anthropic activities [12,13].

Antibiotic-resistant bacteria have been isolated in several bat species worldwide [14], and it seems that bats are more likely to carry antibiotic-resistant bacteria than other taxonomic wildlife groups [15]. However, despite their potential role as environmental reservoirs and vectors of AMR, no studies have comprehensively evaluated the occurrence of resistant strains in European bat populations.

The levels of resistance reported in the literature vary according to the bacterial species examined and the geographical site of the study [14]. Previous research revealed low AMR levels in bacteria isolated from bats living in well-preserved environments away from human activities [16]. However, little is known about the impact of anthropic activities on AMR levels in bat populations [16].

The purpose of this work was to evaluate the distribution of AMR in strains of Gram-negative and Gram-positive bacteria isolated from six Southern-Italian bat populations living in the territory of Calabria and Sicily by mass spectrometry MALDI-TOF (matrix assisted laser desorption/ionization—time of flight mass spectrometry). Our analysis included a wide range of bacterial species, antibiotic molecules, and different geographic sampling areas, as all these factors are known to influence antibiotic resistance patterns [17].

## 2. Materials and Methods

### 2.1. Strains Used

We studied 413 strains of Gram-negative bacteria (Appendix A) and 183 strains of Gram-positive bacteria (Appendix A) previously isolated from rectal (R), oral (O) and conjunctival (C) swabs of 189 bats belonging to four insectivorous species [*Miniopterus schreibersii* (n. 175 Gram-negative and 95 Gram-positive), *Myotis capaccinii* (15 Gram-negative and 10 Gram-positive), *Myotis myotis* (112 Gram-negative and 44 Gram-positive), and *Rhinolophus hipposideros* (111 Gram-negative and 34 Gram-positive)] (Table 1) [18].

The bats lived in six areas of the Calabrian and Sicilian territories (Southern Italy) (Table 2). All the species examined are commonly found in the Mediterranean basin, normally taking refuge in underground habitats (especially caves). They feed on insects, particularly Trichoptera, Neuroptera, Diptera, and Lepidoptera. *M. capaccinii* is also known to feed on small fish and therefore prefers habitats with water courses in which to hunt. The emergency points from the occupied caves are the same for different bat species; however, once outside the cave, different species occupy different trophic niches.

For comprehensive information on the bacteriological analysis and sampling sites, we refer the reader to Foti et al., 2022 [18].

### 2.2. Antimicrobial Susceptibility Testing

We evaluated the antimicrobial susceptibility of all strains isolated (Appendix A) by a Disk diffusion test [19], using the same protocol we employed in a previous study [20]. We evaluated the susceptibility of Gram-negative strains to 20 antibiotic molecules belonging to 9 different classes and that of Gram-positive strains to 20 molecules belonging to 11 classes of antibiotics. Strains exhibiting resistance to three or more antimicrobial families were classified as multidrug resistant (MDR) [21].

### 2.3. Data Analysis and Visualization

In order to evaluate whether any of the sites or bat species harboured a significantly higher number of multidrug-resistant (MDR) strains than the others, we performed multiple pairwise comparisons using the chi-square ($\chi^{2}$) test. For both tests, we fixed the significance level at α=0.05 and applied the Bonferroni correction for multiple pairwise comparisons [22]. This led to a significance level for each individual hypothesis of α=0.0033 for comparisons between sites (15 pairwise comparisons between six sites) and α=0.0083 for comparisons between species (6 pairwise comparisons between four species), corresponding to critical chi-values of $\chi_{0.0033}=8.634$ and $\chi_{0.0083}=6.968$, respectively.

All statistical tests have been performed using MATLAB_R2016b. Violin plots have been produced using the open-source MATLAB function violin.m [23], choosing a kernel bandwidth bw=6  to perform kernel density estimation using the same function.

## 3. Results

The analysis of antimicrobial susceptibility revealed antimicrobial resistance for the large majority of the tested molecules. In both Gram-negative and Gram-positive strains, the resistance levels ranged from one to thirteen molecules (Appendix A).

### 3.1. Gram-Negative

Gram-negative strains displayed a broad and diverse pattern of resistance to almost all molecules tested, with the exception of ciprofloxacin and enrofloxacin (Figure 1, upper panel). Most of the Gram-negative strains exhibited resistance to colistin sulfate (81.6%), amoxicillin (70.7%) and amoxicillin+clavulanic acid (58.1%); high values were also found for ampicillin (42.9%), streptomycin (40.2%) and minocycline (21.1%). No resistance was detected against fluoroquinolones.

170 strains (41.2%) displayed multidrug resistance (Appendix A). The most abundant MDR patterns recorded were combined resistance to aminoglycosides, penicillins and polymyxins (50 strains) and to penicillins, tetracyclines and polymyxins (24 strains) (Appendix A). The levels of MDR in the most frequently isolated (n>10 strains) gram-negative bacterial genera are shown in the left panel of Figure 2. Strains from *Pseudomonas* (78.9%), *Serratia* (61.5%) and *Providentia* (60.9%) displayed the highest levels of MDR, followed by *Enterobacter* (52.5%) and *Morganella* (33.3%).

### 3.2. Gram-Positive

The gram-positive strains isolated displayed varying levels of resistance to all molecules tested (Figure 1, lower panel). Most of the Gram-positive strains exhibited resistance to ceftazidime (88.3%), oxacillin (73.3%), and cefovecin (68.3%) (lower panel of Figure 1). Only a small percentage of strains displayed resistance to ampicillin + sulbactam (1.7%) and ticarcillin + clavulanic acid (1.1%).

The right panel of Figure 2 shows the levels of MDR found in the most frequently isolated (n>10 strains) gram-positive genera. *Enterococcus* showed the highest levels of MDR (86.2%), followed by *Staphilococcus* (60.3%). The most abundant MDR patterns observed were combined resistance to aminoglycosides, cephalosporins and penicillins; to Aminoglycosides, cephalosporins, lincosamides and penicillins; to aminoglycosides, cephalosporins, carbapenems, lincosamides, macrolides and penicillins (Appendix A).

### 3.3. Comparisons between Sites

The pattern of resistance against individual molecules found at different sites is shown in the left panel of Figure 3, while the right panel (Figure 1B) shows the levels of MDR. The highest levels of MDR were found in Grotta dei Pipistrelli (Pantalica) (77.2%), more than 30% higher than those found at other sites. Pairwise comparisons between sites using the chi-square test revealed that this difference is significant in the cases of Grotta Chiusazza (χ=11.07, p<0.001), Grotta dei Pipistrelli (Cassano) (χ=14.17, $p<10^{-4}$), Grotta del Burrò (χ=18.35, $p<10^{-4}$) and Greve Grubbo (χ=24.48, $p<10^{-6}$), but not in the case of Grotta Chiusazza (χ=2.40, p=0.12).

### 3.4. Comparisons between Bat Species

The levels of resistance against individual molecules and the levels of MDR found in different species are shown in the left and right panels of Figure 4, respectively. While *M. schreibersii* displayed the highest levels of MDR (53.6%), no pairwise comparison between species using the chi-square test indicated any significant differences in MDR levels (p>0.05 for all individual comparisons).

## 4. Discussion

The spread of antimicrobial resistance (AMR) constitutes a major threat to both human and animal health. Its spread can be accelerated by wild animal populations, which acquire resistant bacteria through contact with anthropic environments [16,24,25]. The lifestyle and feeding habits of bats make them particularly likely to acquire and spread resistant bacteria, as indicated by the higher levels of resistant bacteria isolated in bats than in other wild animals in the same territory [26,27]. Given their potential exposure to anthropic sources of contamination, bats should be included among the animal species that can act as reservoirs of antibiotic-resistant strains.

Previous studies revealed profound differences in resistance levels across different bacterial species and geographical areas [17], suggesting that surveys evaluating the environmental impact of antibiotics should be performed on different bacterial species and multiple sampling sites. In this study, we evaluated the occurrence of AMR in 413 Gram-negative and 183 Gram-positive bacterial strains isolated from six bat populations in Southern Italy. Our analysis revealed the pervasive presence of resistant bacteria at all sampling sites (Figure 3), although the resistance levels varied depending on the molecule tested (Figure 1). Grotta dei Pipistrelli (Pantalica) harbored the highest levels of multidrug resistance (77.2% of isolates; Figure 3), a significantly higher proportion than that observed at any other sites (with the exception of Grotta Palombara). On the contrary, pairwise comparisons did not reveal any significant differences in MDR levels across bat species.

The relatively high level of MDR we have found indicates that bats can act as reservoirs for resistant antibiotic strains (Figure 3 and Figure 4), potentially facilitating their diffusion to other animal and human populations. This result contrasts starkly with the low AMR levels found in bat populations living in a South American natural reserve [16]. This difference can be explained by the strong anthropic presence near the six sampling sites of our study, a factor known to have a major impact on AMR levels [16,24,25].

Direct and indirect contact with humans poses a risk for the spread of AMR in and through bats. Benavides et al. [28] found the same blaCTX-M-15 gene in *E. coli* ST isolated from bats and pigs, suggesting the transmission of bacteria between bats and livestock. Different routes of transmission can be hypothesized, for example, the ingestion by bats of food or water contaminated by fecal material from livestock.

The bats we examined fed on insects (e.g., flies) that may come into contact with farm animal droppings containing resistant bacteria. Given the long lifespan of these bat species (up to 30 years), resistant strains can survive for a long time and can be easily transferred inside the colony during periods of hibernation due to their habit of aggregating in crowded roosts [29]. Certain geographical features might be responsible for the anthropogenic spread of AMR. For example, several watercourses flow through Grotta dei Pipistrelli in Pantalica (SR), into which the wastewater of different municipalities is discharged (including that of several hospitals). This can potentially account for the high levels of AMR found at that site (Figure 3).

A higher level of AMR in bacteria isolated from rectal and oral swabs might provide information about the transmission route of resistant bacteria. However, comparisons between resistance levels at different bodily sites (rectal, oral, and conjunctival) did not reveal any significant differences (Appendix A). The hypothesis that the detected resistance is a consequence of exposure of human origin would be supported by the resistance detection against synthetic antibiotics (trimethoprim and nalidixic acid) [17], but it must be considered that such resistance could be non-specific because of mechanisms related to membrane permeability [30].

Another possible transmission route is related to the employment of antibiotics in sheep and cattle farming. It would be insightful to compare the levels of AMR in bacteria isolated from cattle and sheep breeding in the vicinity of the sampling sites with those reported in the present study. Similarities in AMR distribution might be indicative of the spread of resistant bacteria from livestock treated with antibiotics to wild animal populations. Further studies are necessary to assess the impact of antibiotic use and ecological factors, such as the exposure to natural antibiotics produced by other microorganisms [17].

Bats can spread bacteria in the environment and transmit them to humans both directly (in situations of synantropization) and indirectly, by infecting intermediate hosts or contaminating drinking water or raw food [31]. However, despite their potential role in the spread of resistant strains, few studies have evaluated the occurrence of bacterial pathogens in wild bat populations [5,6]. Investigations on the microbial flora in bats on the European continent are even scarcer and often limited to sporadic reports of single pathogens [32,33,34,35,36,37,38,39]. Our results provide comprehensive data on the occurrence of AMR in Southern-Italian bats, indicating their role as environmental reservoirs of resistant strains. Further studies are needed to enhance our understanding of transmission mechanisms between bats, domestic animals and humans in order to develop new control protocols and monitoring strategies.

In line with previous studies, the major antibiotic resistance recorded in Gram-negative bacteria was against penicillins [14]. Our analysis also revealed a high resistance to streptomycin, in agreement with previous results [15,26], and to colistin sulfate, which had previously been found only at lower levels [15,31].

Some geographical features could favor the anthropic distribution of AMR in wild populations. The Grotta dei Pipistrelli of Pantalica opens in a natural area not subject to significant anthropic pressures (absence of crops and grazing within the Reserve). For this reason, despite the complexity of the food web and the foraging habits of the investigated species, we hypothesize that the acquisition of resistant bacteria is mediated by the watercourses flowing within the Reserve, which could intercept discharges coming from urban centers. Wastewater can contain complex mixtures of pharmaceuticals, detergents, and bacteria of human and animal origin [12]. However, further studies are necessary to evaluate whether the prevalence of resistant bacteria is a consequence of human antibiotic use, as AMR levels can also be influenced by other ecological factors, such as exposure to natural antibiotics produced by other microorganisms [17].

Despite the hypotheses formulated, it remains impossible to define with certainty the source of transmission of antibiotic-resistant bacteria in wildlife. Vittecoq et al., in a 2016 review, compared the results of a number of studies on the subject, declaring that they were unable to draw any statistically significant conclusions about the primary source of transmission of AMR, which therefore remains undetermined [40]. Nevertheless, our results prompt us to recommend a more prudent use of antimicrobials in humans and animals. The finding of resistance against antibiotic molecules intended for human use only, such as carbapenems, is worrying, as is the high resistance to colistin. Polymyxins, such as colistin and polymyxin B, are currently used as last-resort antibiotics in the treatment of human infections caused by multidrug-resistant Gram-negative bacteria. In view of this and in light of the increase in bacterial resistance to this molecule, the European Medicines Agency (EMA) has recommended that colistin-containing medicines should only be used as a second line treatment in animals and has decreed a gradual decrease of its use in livestock by setting a threshold for use that should be limited to a maximum of 5 mg/PCU (population correction unit). This limit should lead to a 65% reduction in use at the European level. Member States were invited to tighten further by bringing the threshold to 1 mg/PCU in a short time. In light of the results of this study, we recommend the development of more stringent surveillance measures to restrict the excessive use of such antibiotic molecules in farm animals and, by doing so, limit the spread of AMR to wild animal populations.

## 5. Conclusions

Bats can spread bacteria in the environment and transmit them directly to humans in situations of synanthropization and indirectly by either infecting intermediate hosts or through contamination of drinking water or raw food [31].

Geographical location plays an important role in determining multidrug resistance (MDR) levels in bat populations, although no significant differences were revealed across different bat species. Anthropic contamination might have a strong impact on the spread and rise of MDR. Further studies are needed to increase our knowledge of human-bat interactions in order to develop more effective surveillance protocols. Surveillance should above all be aimed at controlling bats that frequent environmental interfaces at risk of close contact with humans where the presence of antibiotic residues or co-selecting agents, such as heavy metals, can be hypothesized.

Finally, with a view to surveillance and containment of the antibiotic resistance phenomenon, we underline again the importance of an appropriate use of antimicrobials in relation to the purposes of the case and being bound by a responsible choice of healthcare professionals.

## Figures and Tables

**Figure 1 animals-13-00966-f001:**
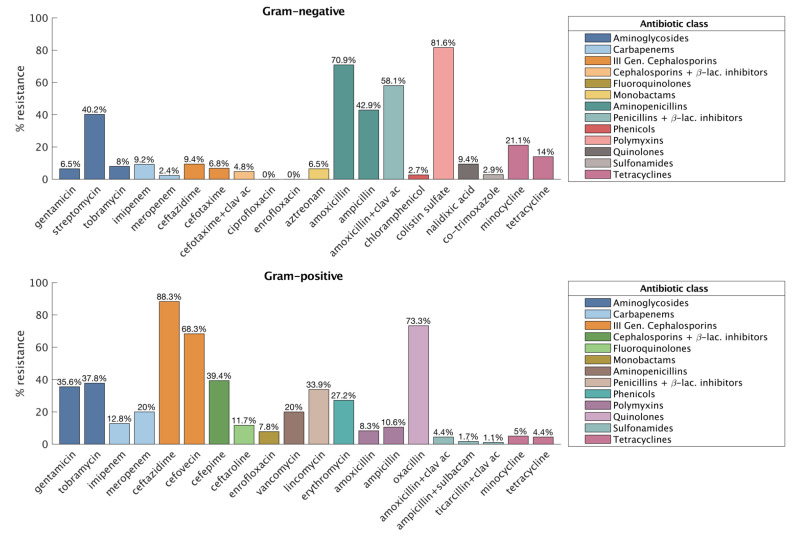
Percentage of gram-negative (upper panel) and gram-positive (lower panel) strains resistant to specific molecules.

**Figure 2 animals-13-00966-f002:**
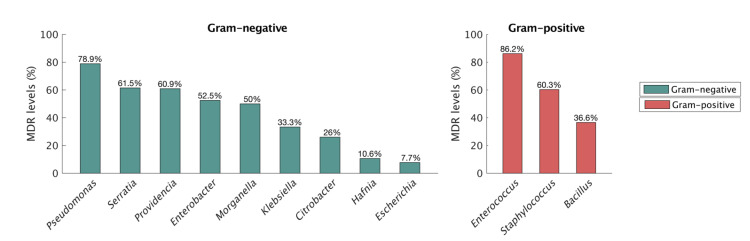
Percentage of multi-drug-resistant (MDR) strains in Gram-negative (**left** panel) and Gram-positive (**right** panel) bacterial genera.

**Figure 3 animals-13-00966-f003:**
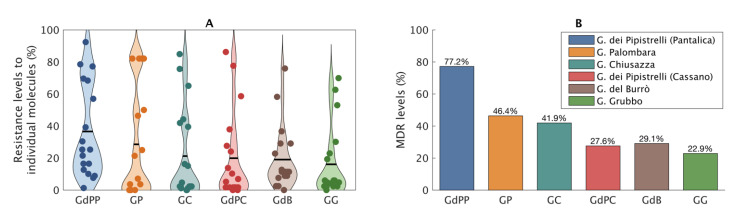
(**A**) Resistance levels to individual molecules and (**B**) percentage of multi-drug resistant (MDR) strains at different sites. Figure legend: GdPP = Grotta dei Pipistrelli (Pantalica), GP = Grotta Palombara, GC = Grotta Chiusazza, GdPC = Grotta dei Pipistrelli (Cassano), GdB = Grotta del Burrò, GG = Greve Grubbo.

**Figure 4 animals-13-00966-f004:**
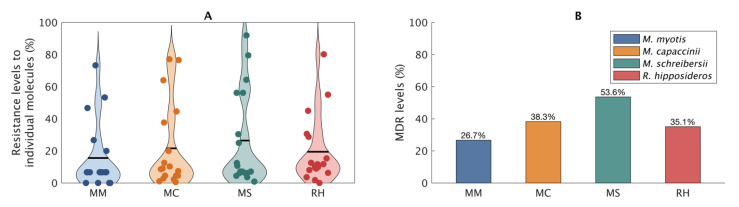
(**A**) Resistance levels to individual molecules and (**B**) percentage of multi-drug resistant (MDR) strains in different bat species. Figure legend: MM = *M. myotis*, MC = *M. capaccinii*, MS = *M. schreibersii*, RH = *R. hipposideros*.

**Table 1 animals-13-00966-t001:** Individuals sampled.

Superfamily	Family	Subfamily	Species	n. Individuals
Vespertilionoidea	Vespertilionoidae	Myotinae	*Myotis myotis*	47
			*Myotis capaccinii*	8
	Miniopteridae		*Miniopterus schreibersii*	91
Rhinolophoidea	Rhinolophidae		*Rhinolophus hipposideros*	43
Total				189

**Table 2 animals-13-00966-t002:** Number of sampled individuals in the six study areas by species.

Site	Number of Sampled Individuals				
	*Myotis myotis*	*Miniopterus schreibersii*	*Rhinolophus hipposideros*	*Myotis capaccinii*	Total
Grotta dei Pipistrelli Pantalica (SR)	13	12	17	8	50
Grotta Palombara (SR)	5	9			14
Grotta Chiusazza (SR)	12	13	7		32
Grotta dei Pipistrelli Cassano (CS)	16	15			31
Grotta del Burrò (CT)	1	15	17		33
Grave Grubbo (KR)		27	2		29
Total	47	91	43	8	189

## Data Availability

The data that support this study are available in the Mendeley Data repository at doi:10.17632/tct4zh9fy6.1.

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
