# Peer review of "Antimicrobial Resistance in Physiological and Potentially Pathogenic Bacteria Isolated in Southern Italian Bats"

_animals, 2023, doi:10.3390/ani13060966_

Round 1

Reviewer 1 Report

Comments to the Author

This study was to evaluate the levels of drug resistance among 413 Gram-negative and 183 Gram-positive bacteria, previously isolated in six bat populations living in Sicilian and Calabrian territory (Italy), using the Disk diffusion method. Antimicrobial susceptibility analysis showed high resistance to some of the molecules tested and the presence of numerous multi-drug resistant (MDR) strains. It has some scientific significance but is not clearly articulated and needs to be refined.

Other comments

Q1:  Line73: Please supplement the strain identification method, mass spectrometry or16s?

Q2:  Line 91: “Sampling sites” Please represent this in a map, administrative region, which should give a better presentation and a better indication of the research implications.

Q3:  How about antibiotic usage in local farming industries? How about local farm or wild animals carrying drug-resistant strains? How is it related to the resistant strains carried by the bats in this study? Some more details that can be discussed.

Q4:  There are also some punctuation marks in the article, case writing, labeling inaccuracies, please revise it seriously

Author Response

Dear Sir/Madam,

Many thanks for the useful comments and insights. We agree with your suggestions and have provided to correct the manuscript according to your instructions.

We report below, point by point, the description of the revisions to the manuscript and the responses to your comments.

Q1: Line73: Please supplement the strain identification method, mass spectrometry or16s?

We included more information about the strain identification method.

Furthermore, in the text we refer to a previous study for additional information (“for Comprehensive Information on the Bacteriological Analysis and Sampling Sites, we refer the reader to Foti et al., 2022 [18]”)

Q2:  Line 91: “Sampling sites” Please represent this in a map, administrative region, which should give a better presentation and a better indication of the research implications.

To reduce the similarity with a previous study, we have eliminated the Sampling sites paragraphs. The information about sampling sites can be found in Foti et al., 2022 [18].

Q3:  How about antibiotic usage in local farming industries? How about local farm or wild animals carrying drug-resistant strains? How is it related to the resistant strains carried by the bats in this study? Some more details that can be discussed.

We are grateful to the reviewer for these insightful comments. We have elaborated on this in the discussion.

Q4:  There are also some punctuation marks in the article, case writing, labeling inaccuracies, please revise it seriously

We have revised the manuscript.

Best wishes,

Dr. Maria Foti

Reviewer 2 Report

Authors present an interesting study on antibiotic resistance in wildlife species of interest (bats) regarding infectious diseases. The manuscript is well-written and the results are easy to understand and interpret, since the authors chose the right amount of information to present in the article and supplementary materials. In my opinion, authors should receive credit for their work. 

However, I have some comments and suggestions that I believe will improve their manuscript substantially. 

1. Please re-check the instructions for authors because I think the abstract should not present the subtitles (as "background"), although should be organized according to those subsections.

2. Please rephrase this sentence (line 39) "The recent Covid-19 pandemic demonstrated how bats can act as a reservoir and vehicle for dangerous infectious agents". I  understand what you mean with it, because bats are pointed out as one of the possible hosts involved in COVID19 crossing species barrier. However, it is not 100% proven, and I believe they were not the only species involved in the process. Therefore, or you cleary mention an article that supports this theory or you present this idea with some reasonable doubt.

3. Line 68-72 (but this comment is valid for the rest of the manuscript). Sometimes you refer to your results using past verb forms, others you use the present. I am not an English speaker, but I normally use the past when I write an article. Nevertheless, you should choose one of these forms of writing and be coherent in the whole manuscript. Please revise this situation.

4. Something is not right in the text formatting. It changes between the Methodology and Results sections. Please check this.

5. We collected samples using "rectal (R), oral (O) and conjunctival (C) swabs". I believe it would be relevant to say something in the Discussion regarding the differences (type of strains, bacteria abundance/diversity, resistance...) you found between those samples. The route of contact and transmission of these bacteria can provide useful information regarding the "origin of the problem". What do you think?

I have nothing further to add.

Author Response

Dear Sir/Madam,

Many thanks for the useful comments and insights. We agree with your suggestions and have provided to correct the manuscript according to your instructions.

We report below, point by point, the description of the revisions to the manuscript and the responses to your comments.

Please re-check the instructions for authors because I think the abstract should not present the subtitles (as "background"), although should be organized according to those subsections.

We have eliminated the subtitles.

Please rephrase this sentence (line 39) "The recent Covid-19 pandemic demonstrated how bats can act as a reservoir and vehicle for dangerous infectious agents". I understand what you mean with it, because bats are pointed out as one of the possible hosts involved in COVID19 crossing species barrier. However, it is not 100% proven, and I believe they were not the only species involved in the process. Therefore, or you cleary mention an article that supports this theory or you present this idea with some reasonable doubt.

We have rephrased this sentence.

Line 68-72 (but this comment is valid for the rest of the manuscript). Sometimes you refer to your results using past verb forms, others you use the present. I am not an English speaker, but I normally use the past when I write an article. Nevertheless, you should choose one of these forms of writing and be coherent in the whole manuscript. Please revise this situation.

We corrected this.

Something is not right in the text formatting. It changes between the Methodology and Results sections. Please check this.

We checked

We collected samples using "rectal (R), oral (O) and conjunctival (C) swabs". I believe it would be relevant to say something in the Discussion regarding the differences (type of strains, bacteria abundance/diversity, resistance...) you found between those samples. The route of contact and transmission of these bacteria can provide useful information regarding the "origin of the problem". What do you think?

We thank for the relevant suggestion. Unfortunately, we have found no significant difference in the resistance levels across these sampling locations. We have added two sentences in the Discussion to explain this. Other differences (types of strains, abundance/diversity patterns, etc…) have already been discussed in a previous study (Foti et al., 2022 [18]).

Best wishes,

Dr. Maria Foti
